# Pressure Sensors: Working Principles of Static and Dynamic Calibration

**DOI:** 10.3390/s24020629

**Published:** 2024-01-19

**Authors:** José Dias Pereira

**Affiliations:** 1ESTSetúbal, Instituto Politécnico de Setúbal, Rua do Vale de Chaves, Estefanilha, 2910-761 Setúbal, Portugal; dias.pereira@estsetubal.ips.pt; 2Instituto de Telecomunicações, 3810-193 Aveiro, Portugal

**Keywords:** sensor, pressure measurement, static and dynamic calibration, measurement errors

## Abstract

This paper starts with an overview of the main principles used for pressure measurements, focusing on their usage in industrial applications’ domains. Then, the importance of calibration procedures, namely, static and dynamic calibration of pressure sensors, is analyzed. Regarding calibration, it is important to note that there are several applications where the pressure signals to be measured can have large variations in short periods of time. In industrial applications, particularly in continuous production processes, generally, dynamic pressure measurements are less common; however, they are still required in several cases, such as control loops that are very sensitive to pressure variations, even if the frequencies of those variations are in the range of a few tens of hertz, or even lower. The last part of the paper presents the hardware and software of a flexible and low-cost static and dynamic pressure calibrator that also presents the capability to generate arbitrary waveform pressure signals for calibration and testing purposes. The proposed calibrator also includes the following advantages: remote pressure sensing capabilities that can be used to minimize calibration errors, such as those associated with capillary effects and pressure leakages; portability; and low cost. The paper ends with some experimental results obtained with the proposed calibrator.

## 1. Introduction

Pressure measurement has been a point of interest in science for many years, and the research interest in this area has continued in recent decades up to the present. With the advent of digital technology, a huge variety of equipment has spread across the market in various applications. One issue that is particularly important in the instrumentation and measurement areas, with a huge impact on industrial applications, is calibration. In industrial applications, there are a large number of cases where static calibration of pressure measurement systems seems to be enough, but there are some important exceptions where dynamic calibration is essential. For example, blade breakages, which are common faults in centrifugal pumps, cause flow and pressure pulsations that can be detected by pressure measurements [1]. Thus, in order to accurately detect blade breakages, pressure pulsation monitoring in the output of the pumps and the measurement of output flow variations are important to implement condition-based maintenance of these devices. Another example, in the same area, is related to pressure pulsations generated in the headbox feed pump of a paper machine [2] that can cause unacceptable variations in paper basis weight in the machine direction. Thus, in pressurized paper machine headboxes, pressure measurements must be very accurate and the dynamic characteristics of the pressure measuring chain must be known. Dynamic pressure measurements are also required in industrial deaerators that are used to remove dissolved gases, such as oxygen, nitrogen and carbon dioxide, from water to minimize corrosion risks [3]. On the other hand, in other application areas, such as military engineering, chemical explosion tests [4], petroleum exploration and well testing [5], gas turbine testing and combustion engine testing [6,7], and biomedical applications [8], among others, dynamic calibration of pressure sensors is essential. A typical example that can be referred to in the automotive area is related to air flow control in combustion engines [9]. Pollution of the combustion air can result in a reduction in the oxygen available for combustion and can alter the chemistry of exhaust emissions, with a negative impact on environmental air quality. Thus, several applications in different domains that use pressure sensing devices require the dynamic characterization of the pressure acquisition channel and not only of the pressure sensors that are part of it. In these cases, it is necessary to measure dynamic pressure signals with different waveforms, amplitudes and frequency contents. This requires that the pressure sensors used for these purposes be judiciously selected in terms of their natural frequency, rise times and other sensitive dynamic parameters. Furthermore, it is important to underline that, regarding dynamic calibration, capillary, tubing and electronic dynamic responses must be considered together to evaluate the dynamic behavior of the complete pressure measuring channel. Considering, as an example, only capillary effects, according to [10,11,12,13], when high-frequency dynamic pressures are to be measured, there are many factors that should be considered. Primary among them are tubing length and tubing diameter. Additionally, temperature, the internal volume of the transducer, the geometric layout of the tubing and the type of tubing must also be considered. When the pressure waves reach the sensing device, there are two main sources of distortions. The pressure waves could be shifted, altering existing time correlations, or the magnitude of the pressure waves could be distorted, giving erroneous measurements. For example, if a resonance exists in a capillary, false data could be measured, masking true values of pressure-related quantities and giving, potentially, very large measurement errors. The final solution for eliminating errors generated by pneumatic systems is to individually calibrate and tune the systems before use. This task requires the usage of well-known signal frequencies to supply the different tubulations and then tuning each one to remove any resonance present and to minimize phase shifts or amplitude distortions. Finally, it is important to note that there is no standardized method to quantify the frequency response of a pressure transducer, and several research activities are still being developed in this area [14,15], there having been several calibration solutions proposed over time. Table 1 summarizes, in comparative terms and in a qualitative way, the advantages and disadvantages of some of those solutions for pressure calibration purposes.

From the previous table, it can be underlined that the main novelties associated with the proposed prototype are its capability to minimize calibration errors due to its remote sense compensation capability and its larger pressure dynamic range, besides its capability to generate arbitrary waveform pressure signals.

## 2. Pressure Measuring Principles

This section includes an overview of the different measurement principles that are used in pressure sensors. Some of the main characteristics associated with the different measuring principles are underlined and typical responses times associated with the different measurement principles are mentioned. Regarding this last characteristic, which is directly related to the dynamic behavior of the sensing devices, it can be noted that the typical response times of pressure sensors can be as low as 1us. Newer implementation techniques based on MEMS are pushing these response times even lower.

### 2.1. Bourdon Tube, Bellow, Diaphragm and Capsule

These measurement principles are based on the mechanical and elastic deformation of different elements that sense pressure amplitudes. One of the oldest, but still used, devices of this type is the Bourdon manometer, also known as the Bourdon tube, which consists of a C-shaped elastic metal tube closed at one end and with an elliptical cross section. The materials that are used to build these instruments are, typically, stainless steel, phosphor bronze and copper/beryllium, due to their elastic characteristics, and it is possible to achieve, with these instruments, repeatability values better than 0.1% of the full scale and measuring ranges between 10 kPa and 100 MPa with an accuracy of almost ±1% of the full scale. As the main advantages of these instruments, their high robustness, which is essential for working in harsh environments, and their large measuring ranges must be underlined. Another measuring element, used frequently, is the diaphragm pressure gauge. In this case, the deflection of a thin and flexible membrane is used to measure, usually, fluid pressures. The deformation of the diaphragm pressure sensor is, typically, measured by piezoresistive, capacitive or inductive sensors. In the piezoresistive case, the applied pressure is transmitted to a thin-film piezoresistive material, deposited on the diaphragm surface, whose resistance varies with the applied pressure. In the capacitive case, the diaphragm is between two fixed capacitive plates, forming a dual variable capacitor. In this case, it is easily possible to measure relative, absolute and differential pressures with the same sensor. A typical signal conditioning circuit to measure the dual capacitive variations is based on the Sauty bridge [19]. In the inductive case, the diaphragm is placed between two coils that form an inductive circuit. The diaphragm’s deflection changes the inductance of the circuit, generating an alternate voltage that depends on pressure values. It is important to note that an intrinsic advantage of diaphragms is that they isolate internal components of the instrument from the media, making it suitable for corrosive or contaminated liquids or gases. The pressure ranges of diaphragm gauges fall between 1 kPa and 4 MPa. When pressure ranges are low, in the order of a few kilopascals, diaphragms with a larger diameter are used in order to increase the sensitivity and accuracy. Regarding measurement accuracy, diaphragm pressure gauges reach accuracies as low as 0.5 percent of their full scale. For low pressures of about 1 kPa, the diaphragm gauge works in its limits of sensitivity. For this range of pressures, the diaphragm should be very thin in order to have enough elasticity, and the usage of capsules that are made up of two diaphragms is a more suitable solution for sensitive pressure measurements, since the sensitivity will be twice as large as that delivered by a diaphragm of the same diameter. However, since capsules are not self-draining, they cannot be used for liquids and are only used for the measurement of gas pressures. Considering that the indicators used in this type of sensor is based on mechanical devices, the response time is in the order of hundreds of milliseconds. For example, the response time of a Bourdon tube is typically about 0.2 s. Based on the definition of pressure, which is a scalar quantity defined as the modulus of the applied force divided by the units of area, Figure 1 represents a deadweight tester [20] that enables static calibration of pressure manometers with an accuracy better than 0.1% for readings and an operating pressure that can vary between 103.4 and 689.5 kPa.

### 2.2. Piezoresistance and Strain Gauge

These elements of the resistive type case be used directly or indirectly with deformable mechanical structures to perform pressure measurements. Piezoresistivity refers to the change in electrical resistance with deformation/contraction as a result of an applied pressure. The vast majority of piezoresistances are made up of crystalline elements (strain gauges). In turn, strain gauges are typically of metallic types that have a strain gauge sensitivity around 2, but, in some applications, they can be made of semiconductors, the sensitivity being much higher. However, the usage of metallic strain gauges implies special care regarding temperature sensitivity, since resistance variations are too small, in the range of tens or a few hundreds of milliohms (mΩ). For this reason, temperature compensation for strain gauge resistance variations is essential in order to obtain accurate measurements. Conditioning circuits for these sensing elements are usually based on Wheatstone bridges that provide differential output voltages, with almost null common voltage values, zero adjustment capability and an easy way to implement temperature compensation by using active and passive strain gauges or multiple strain gauges connected in the same Wheatstone bridge. It is important to underline that, nowadays, it is common to use so-called thin-film technologies, where resistive sensing devices are built by using vapor deposition or injection techniques directly into the diaphragm, it being possible to minimize errors caused by the usage of adhesives in the alloys of the so-called bonded wire models. Several techniques based on the manufacture of piezoresistive silicon sensors (silicon substrates) are emerging, but they are susceptible to degradation of their signals as a function of temperature and require complicated circuits for compensation, error minimization and zero calibration. In terms of sensitivity, to maximize the value of this parameter, the strain gauge should ideally be placed in the region of greatest positive and negative strain or stress. The form, location and length of the measuring grid can be optimized, since the precise strain gradient and strain distribution in the measuring body are known at the time of the pressure transducer’s design [21]. Regarding response times, typical values of this measuring principle range from 1 to 10 ms.

### 2.3. Capacitor

Capacity variations caused by pressure variation are also frequently used in pressure sensors. In this case, the pressure applied to a flexible diaphragm causes a variation in the capacitance between the diaphragm’s two metal fixed plates. Typically, the metal plates are very close to each other, in parallel, and the flexible metallic diaphragm is in the middle of the fixed metal plates. This arrangement is known as a differential pressure (d/P) capacity cell, an insulating dielectric, for example, oil glycerin or silicone, being used as a pressure-transmitting medium. Figure 2 depicts a general view of a common industrial differential pressure transmitter and a detailed view of its d/P capacitive cell. Regarding signal conditioning techniques, capacitance variations are sometimes used to vary the frequency of an oscillator or as an element of a capacitor bridge. The variations in the capacitance of the d/P cell are usually used to vary the frequency of an oscillator, taking advantage of the noise immunity of a frequency-coded signal and its easy analog-to-digital conversion (ADC), without requiring a traditional ADC [22]. The frequency or time modulated signals, also known as quasi-digital signals [23], are easily converted to the digital domain, and this contributes to accuracy improvement and to the elimination of drifts embedded in traditional analog/digital converters. The main advantages of capacitive pressure sensing techniques include their large measurement range, high sensitivity, common mode error compensation, accuracy and reliability, among others. This type of sensing technique also includes ceramic measuring cells that are dry measuring cells that operate without oil as a pressure-transmitting medium. The main advantages of this type of cell are the absence of contamination risks, which are ever-present in closed pressurized manufacturing processes; their suitability for applications with extreme temperatures that can range between −70 °C and 400 °C, using remote seal process connections; vacuum measurement capabilities; and very low measuring ranges, as low as from 0 to 50 Pa [24]. Moreover, it is important to say that where there is no oil, no air can get trapped in the system, which can cause drift and large measurement errors. Regarding response time, typical values of this measuring principle can be as low as 500 μs.

### 2.4. Reluctance

This type of pressure sensing device is based on inductance variations caused by pressure variations. As an example, Figure 3 represents a schematic diagram of the measuring principle of a reluctance pressure transducer. The sensor includes a pressure sensing diaphragm and a two-coil inductive half-bridge. The coils are wired in series and are mounted so their axes are normal to the plane of the diaphragm. When a differential pressure is applied to the sensor, the diaphragm deflects away from one coil and towards the opposite coil. The diaphragm material is magnetically permeable, and its presence nearer the one coil increases the magnetic flux density around the coil. Thus, inductance and impedance variations of oppositive effect are sensed by each coil and the change in coil impedances brings the half-bridge out of balance, and a small AC signal appears on the output signal line. Regarding AC excitation signals, the inductive half-bridge of the reluctance sensors is powered by an alternating voltage source whose frequency is, typically, between 1 kHz and 10 kHz and whose amplitude is in the order of a few r.m.s. volts. 

In this type of sensor, whose working principle recalls the working principle of the LVDT, the output signal is quantified as a ratio of the excitation voltage, typically in mV/V of the excitation voltage, when the sensor’s full-scale pressure is applied. Another characteristic of this type of sensor is that the total displacement over a full-scale pressure excursion is very low, in the order of a few micrometers, and there are no mechanical linkages or hydraulics inside the sensor to slow down the sensing element, these sensors being typically over-damped at their natural frequencies.

### 2.5. Piezoelectrics

The piezoelectric material is a crystal that produces a differential voltage proportional to the pressure applied to it on its faces. Some classes of materials, such as quartz, Rochelle salt, barium titanium and tourmaline, among others, accumulate electrical charges in certain areas of their crystalline structures when they undergo physical deformation due to pressure. The relationship between the electrical charge and the pressure applied to the crystal is practically linear, the following being valid as an approximation:$$q={\;S}_{q}\cdot A\cdot p$$
where q, *S_q_*, A and p represent the electrical charge, the sensitivity, the electrode area and the applied pressure, respectively. Other advantages of these sensing devices include their fast response to abrupt pressure variations and their very resonant frequency values. Among the disadvantages, one of the main disadvantages is the required high impedance of tire conditioning circuits, generally provided by charge amplifiers; the high gain amplification required; their susceptibility to noise; and their inability to work with very-low-frequency signals, including static pressure signals. Regarding response time, typical values of this measuring principle can achieve values of about 10 μs, but, as noted, static pressures cannot be measured.

### 2.6. Resonance

The core of the resonant pressure sensor is generally a silicon structure that can resonate or vibrate at a specific frequency. This structure is called a “resonator”. When external pressure is applied to the sensor, it causes a change in the shape or position of the resonator, which in turn changes its natural resonant frequency. This change in frequency is detected and converted into an electrical signal, which can be interpreted as a measurement of external pressure. One interesting implementation of this type of sensor is known as the TERPS (Trench Etched Resonant Pressure Sensor), which is an MEMS (microelectromechanical system)-based resonating silicon sensor that takes full advantage of the beneficial mechanical properties of pure silicon to maximize pressure measurement performance. The mechanical properties of silicon are only changed marginally by changing temperature. This makes temperature correction easy, making it possible to obtain a precision of 0.01% FS (100 ppm). Figure 4 represents the internal structure of a TERSP resonator that includes three distinct layers, namely, a diaphragm, a resonator and a capacitive layer. Normally, the resonator and the diaphragm are made from the same piece of silicon. Regarding response time, typical values of this measuring principle can reach values of about 1 μs. 

### 2.7. Optics

Pressure measuring principles also include optics techniques. A common technique that is used in sensors is the Fabry–Perot interferometer. This technique is generally used to measure wavelengths with high precision, where, essentially, two partially reflecting mirrors (glass or quartz) are aligned and the contrast of maximum fringes and the distance between them are due to mechanical variation. If the distance variations are caused by pressure variations, measurement results can be expressed in pressure units. Figure 5 represents a possible arrangement to build an optical pressure sensor based on light intensity. In this case, the diaphragm is deflected by the applied pressure and then there is an LED light source and two photodiodes. The upper one works as a reference and the lower one is used for pressure measurement, the detected light intensity being dependent on the pressure intensity. This kind of arrangement, using a reference and a measurement photodiode, enables the cancelation of errors caused by common errors that affect, in equal ways, both photodiodes, such as errors caused by temperature variations.

An optical pressure sensor must compensate for the aging of the LED light source by means of a reference diode, which is never blocked by the vane. This reference diode also compensates the signal for build-up of dirt or other coating materials on the optical surfaces. Moreover, because the amount of movement required to make the measurement is very small, hysteresis and repeatability errors are negligible. Other advantages of this type of pressure sensor that are particularly important in some industrial applications are its long transmission distance and the low chemical reactivity of the material, which is ideal for operating in environments with a risk of explosion, in addition to being intrinsically safe [25]. Regarding response time, typical values of this measuring principle can reach values of about 100 μs. 

## 3. Calibration and Pressure Generators

Pressure measurements are generally critical in terms of the quality and efficiency of process control systems. Proper pressure instrument calibration requires the coordination of several factors to ensure precise accuracy. Improving the pressure calibration process can really boost an instrumentation team’s productivity and overall production. However, regarding the need for static or dynamic calibration, it is also important to note that for fast-changing pressure dynamics, pressure sensors with shorter response times are better for accurate measurements, but, when pressure changes are slow, a higher response time can be more adequate due to pressure sensors’ lower cost and greater durability. As always, the best choice depends on the specific application needs. Pressure generators for dynamic calibration can be classified in two main groups: aperiodic and periodic. The former is generally used for calibration methods in the time domain and the latter for calibration methods in the frequency domain. 

### 3.1. Static Calibration

The pressure is said to be static when it remains constant for a significant amount of time, generally during a complete measurement. Figure 6 represents a schematic diagram of a typical deadweight tester for static calibration purposes. A pressure balance includes a piston mounted in a cylinder. The internal pressure that must be generated to support the weight of the rotating piston plus the weight of the reference masses placed over it means that it is possible with this method, under controlled environmental conditions, to achieve an expanded relative uncertainty of some tens of parts per million.

It is important to underline that, in contrast to dynamic pressure calibration, the area of static pressure calibration is well developed and there are a large number of laboratory and portable calibrators used for maintenance purposes of pressure measurement devices.

### 3.2. Dynamic Calibration

Pressure is said to be dynamic when it varies significantly in a short period of time. In this case, what is sought for is not a single time-invariant value of pressure, but rather a time-dependent pressure function. It is important to note that, at present, there are still no traceable dynamic pressure calibration standards, except for sound pressure. In order to quantify the measurement errors associated with using static calibration for pressure sensors that work with time-variable pressures, Figure 7 represents those errors as a function of the normalized signal frequency for three different damping rate values (ζ), namely, ζ = 1, ζ = 0.5 and ζ = 0.25, from the lower to the upper curves, respectively. As a conclusion, for a relative error criterion lower than 5%, the normalized frequencies (ω/ωn) must be lower than 0.23, 0.32 and 0.24, respectively, which means that they must be about 20% of the natural frequency of the transducer.

### 3.3. Pressure Measurement Channel and Transmitter Damping

For faster pressure transducers, such as the pressure sensor DP 15 [26], the flat frequency parameter is the more accurate way to describe the pressure transducer frequency response. The flat frequency is the maximum frequency, in Hz, that the pressure sensor can pass into its signal without significant distortion, typically with a relative measurement error lower than 5%. The most important factors that determine the flat frequency of a pressure measuring channel include the capillary tubes, which interconnect the remote seal diaphragm and the pressure transmitter; the fluid medium inside the capillary; the pressure transducer itself; the transmitter conditioning circuits; and the associated data processing parameters, these being some of the most important configured damping parameters. Regarding pneumatic or capillary tubing, it is important to note that the behavior of the pressure measurement channel is also affected by fluid compressibility effects, temperature oscillations, vibrations and other factors that affect sound propagation inside tubing, including sound speed. Concerning transmitter damping, it is important to underline that the transmitter damping parameter is generally used to parametrize the transmitter and has a direct impact on the dynamic response of the pressure measurement channel, since the dynamic response time is approximately five times the damping time for a 99.3% final value criterion. Damping reduces the impacts of process and electrical noise that affect the output signal of the transmitter. Thus, smart pressure transmitters include, generally, an electronic damping capability that is used to increase the response time of the transmitter and to smooth its output when there are rapid input variations. However, high damping values filter process noise and increase the response time of the pressure measurement channel. On the other hand, low damping values reduce response time but process dynamic behavior, including noise, can be detected, whose measurement can also be relevant to detecting abnormal process behaviors. Regarding dynamic calibration of the complete pressure measurement channel, it is recommended that damping is deactivated or, if this is not possible, that the shortest damping time is selected. 

### 3.4. Periodic Pressure Generators 

In the case of periodic generators, a sine pressure signal of known frequency is applied to the transducer to be calibrated. The steady-state response of the sensor for a series of frequencies can be obtained and thus the frequency response curve can directly be obtained. These generators are particularly suited for the calibration of sensors designed for small pressure amplitudes and low frequencies. However, these limits are being successively surpassed by technological evolution, and there are several applications that include, for example, biomedical instrument testing where the pressure amplitude is lower than a few hundred kilopascals, sometimes much lower, and generating pressure signals with a specific signature is of great importance. This is the case with non-nutritive sucking (NNS), regarding which certain authors have successfully developed several pressure generator prototypes [27,28]. Regarding periodic generators, the best-known solutions are of the cavity type, which can operate in a resonating or a non-resonating mode, and the liquid column generator, the operation principle of this pressure generator being based on a column of liquid that is submitted to sine function acceleration [29]. Regarding the speed of execution, depending on the system that is used, the pressure test can take quite a long time, since several sinusoidal signals with different frequencies, one at a time, are used to obtain the amplitude and phase response of the pressure system under test. 

### 3.5. Aperiodic Pressure Generators 

According to theory, one of the advantages of this type of dynamic pressure test or calibration is that the test is performed with a single pressure stimulus with a step or impulse waveform. However, this type of test works with pressure amplitudes of several megapascals, and it is common to use shock tubes, fast opening devices and dropping weights, among other items. Moreover, when converting the aperiodic time-based results to the frequency domain, high measurement uncertainty has to be accepted. Future improvements regarding aperiodic pressure calibrations must be made using modern control and software techniques, and more effort must also be devoted to including the influence of acceleration and temperature on the behavior of pressure transducers. 

## 4. Proposed Calibration Prototype

The next paragraphs include the hardware and software description of a low-cost, flexible and versatile pressure calibrator that can be used as a periodic or aperiodic pressure generator. Basically, and if required, it can generate any arbitrary pressure waveform that is previously defined by the user, and this advantage is particularly interesting with respect to testing and calibrating pressure measurement devices for biomedical applications. Obviously, beyond the previous important advantages, its accuracy is not so high as the commercial and specific dynamic calibrators [30,31,32] that are tailored for time or frequency dynamic calibration, using aperiodic or periodic pressure generator techniques, but not both techniques, simultaneously in the same calibrator.

### 4.1. Hardware

The block diagram of the electropneumatic pressure regulator (EPPR), used to generate pressure signals for static and dynamic calibration, includes: a pneumatic pressure transducer (PT1); a miniature electro-valve (VENT), used to accelerate the pneumatic discharge of air and for zero calibration purposes; a miniature proportional valve (MPV); and some elementary signal conditioning circuits, namely, a comparator and a voltage follower. Figure 8 represents the EPPR block diagram. The “setpoint” signal, used to define the pressure to be applied to the device under test (DUT), is generated by a multifunction data acquisition board (NI-USB-6002) with 12-bit resolution, a maximum sampling rate of 50 kS/s and a variable DAC output voltage ranging between 0 and 5 V. The two inputs and three outputs of the EPPR block are, respectively, the control voltage (“setpoint”) of the pressure EPPR output, whose value varies between 0 and 5 V; the air supply (AS); the output pressure monitoring signal (MPS), also variable between 0 and 5 V; the air exhaust outlet (VENT); and the interconnection terminal to the device under test/calibration (DUT). Figure 9 represents the block diagram of the calibration system that includes the following elements: the EPPR, which consists of a proportional valve controlled by a variable electrical voltage between 0 and 5 V and which has the ability to monitor the pressure at its outlet through an internal pressure sensor (S.P.); a second pressure sensor that measures the pressure value applied to the remote device (DUT) that is intended to be calibrated and that has a certain output quantity (GSOUT), usually a normalized signal in voltage (1–5 V) or current (4–20 mA); a comparator element (Σ) that compares the pressure values at the EPPR outlet with that existing in the remote device subject to calibration; a PID controller whose output signal, null in the case of equal pressures, is added to the desired pressure “setpoint”; and a computer with a multifunction board (PC + DAQ) that allows the generation of calibration signals, with the amplitude, frequency and form of the waves being adjustable by the technician who performs the calibration. It is the existence of the feedback loop, consisting of the pressure sensor in the DUT, the adder, the comparator and the PID controller, which allows static and dynamic calibration with the ability to remotely sense the pressure actually applied to the DUT. For each of the calibration pressure values, the operation sequence is as follows: the program installed on the PC calculates the corresponding voltage value and the multifunction board imposes the calibration voltage on the EPPR command input; the EPPR imposes an output pressure value that is a function of the “setpoint” voltage of the electronic control circuit; the monitoring pressure signal (MPS) is compared with the pressure signal existing in the DUT and the difference between these pressure values is compensated by the PID control loop until the pressure applied to the DUT corresponds to that intended to be imposed for a given calibration pressure value; and, finally, the value of the DUT output quantity (GSOUT) is measured for calibration purposes. This sequence is repeated for each of the pressure values contained in the calibration cycle that was previously defined. It is important to note that the calibration system block diagram also includes a three-way solenoidal valve [33] used for tubing pressure discharge.

### 4.2. Software

The software of the instrument was developed in LabVIEW 2020. Regarding the pressure measurement part of the instrument, several routines were development for different purposes: data cleansing; peak detection; and extraction of the main signal parameters, such as amplitude, peaks, burst durations, inter-burst durations and signal spectrum (FFT). As an example, Figure 10 represents a front panel of the calibration prototype for NNS stimulation. In this case, the software routine that was developed includes the following main tasks: configuration of the stimulation signal parameters, data generation and on-line adjustment of the configuration parameters. The software also includes routines for storage of historical data, statistical data processing, data reduction and presentation, and database processing functions. Configuration of the NNS signal includes the number of suction bursts, the number of suctions per burst, the inter-burst pause, the waveform type, and its frequency, amplitude and offset.

Concerning flexibility, the proposed system enables different parameters, such as amplitude, frequency, number of pulses, sucking burst per minute and duty rate between stimulating and measuring periods, to be adjusted on-line according to the user’s needs. In terms of frequency and amplitude limits, the calibration prototype can generate pressure signals whose frequency range varies between 0 and 70 Hz and whose pressure amplitude can vary between 0 and 34.5 kPa.

## 5. Experimental Results

This section includes the experimental results associated with static and dynamic calibration of an industrial pressure transmitter [34]. The influence of the capillaries, which are frequently used to interconnect the pressure tabs to the sensing elements, is also analyzed.

### 5.1. Static and Dynamic Calibration of an Industrial Transmitter

Figure 11 represents the experimental setup that was used for testing and calibration purposes.

Besides the part of the system already described in the previous section, the experimental setup includes the pressure transmitter under test, a pneumatic tube used to study capillary effects, the current loop (4–20 mA) with HART [35] communication capabilities, and a reference pressure calibrator [36], used for pressure measurements on the pressure input port of the transmitter. The main characteristics of the transmitter include: a full-scale (FS) range adjustable from ±550 Pa to ±22 MPa; ±0.25% accuracy of the FS range; temperature compensation; temperature errors lower than the +/−0.5% FS range; a pressure cavity volume of 0.012 cu in; and zero and span calibration using external adjustments. It is also important to note that the pressure transducer accepts both liquids and gases directly at the sensing diaphragm, and there are no internal isolation fluids that negatively affect the transducer’s dynamic behavior or cause excessive temperature drift errors. During all the tests, the output signal from the transducer was directly used without any bandwidth restriction caused by transmitter data processing damping. To study the bandwidth restrictions caused by the capillary, a pneumatic tube (P Tube) interconnects the output of the EPPR calibrator to the input pressure port of the transmitter. Regarding the main characteristics of the reference pressure calibrator, it includes 32 pressure modules from 2.5 kPa to 69 MPa, an accuracy better than 0.05% FS and temperature-compensated operation from 0 °C to 50 °C, besides its supplementary capabilities as an accurate multimeter, frequency and pulse generator, and RTD and thermocouple simulator.

Static calibration tests were performed with the following setup and procedure: the transmitter was set up for relative pressure measurement; the measuring range used was between −34.5 kPa and 34.5 kPa; the sensitivity was equal to 0.232 mA/kPa; and the pressure calibration values were uniformly distributed over the measuring range with increments equal to 1.724 kPa. Figure 12 and Figure 13 represent the static calibration results and the relative errors that were obtained, respectively. The relative errors were calculated against the FS of the transmitter, and its mean, standard deviation and maximum values were equal to 0.3087%, 0.1755% and 0.5824%, respectively. It is important to underline that these errors are a little over the specifications, which specify ±0.25% FS, including non-linearity, hysteresis and non-repeatability errors, but they can be reduced, since all the relative errors are always positive and can be reduced by adjusting the zero of the pressure transmitter.

Using the same setup and procedure that were considered previously for static calibration, a dynamic calibration of the pressure measurement channel was performed using a step-down aperiodic pulse that changed abruptly between 6.895 and 0 kPa. A constant voltage of 1 V was applied to the EPPR device to fix a constant pressure value in the pressure channel, and then at t = 125 ms the fast-opening three-way valve was actuated to depressurize the channel, the pressure oscillation being measured after the valve’s opening. The pressure measurement channel was adjusted for a damping factor (ζ) equal to 0.25. This adjustment was performed by varying the capillary length. Figure 14 and Figure 15 represent the theoretical and experimental dynamic calibration results and associated relative errors, respectively, using normalized amplitude values in the vertical scales of the graphs.

Regarding the relative error results, it is important to note that the magnitude of the relative error was mainly caused by the uncertainty associated with the damping value of the pressure measurement channel that was theoretically considered (ζ = 0.25) and, above all, by the fast-opening limitations of the three-way solenoid valve that was used to generate the pressure step transition. Moreover, assuming as valid a second-order transfer function approximation for the pressure measurement channel, the natural frequency that can be obtained from the experimental results is, approximately, equal to 64 Hz. It is important to note that, using [16], it is possible to estimate the value of the pneumatic tube diameter:(1)$$D=\sqrt{\frac{8\cdot \mu }{f_{n}\cdot \rho \cdot \zeta }}$$
where *μ* represents the dynamic viscosity of the air, approximately equal to 1.81⋅10^−5^ Pa·s; fn represents the natural frequency of the pressure measurement channel that was experimentally obtained, almost 64 Hz; ρ represents the air density, approximately equal to 1.293 kg/m^3^; and ζ is equal to 0.25, as previously noted. Replacing the numerical values (expression numbers), a tube diameter of almost 2.65 mm is obtained. The pneumatic tube that was used has a diameter equal to 2.5 mm, the relative deviation between the real and estimated diameter values being about 6%. This deviation is acceptable, since the difference can be easily justified by the influence of the pressure sensor cavity volume, not considered (expression number), and by the influence of temperature and pressure variations on air density and dynamic viscosity.

### 5.2. Capillary Effects

In the next paragraphs, the bandwidth reduction of a pressure measuring channel, caused by capillary effects, is briefly analyzed. The natural frequency of a capillary tube connected to a pressure sensor depends on the sensor cavity volume and capillary dimensions, these being as approximately given in [37]:(2)$$\omega_{n}=\frac{c}{L\cdot \sqrt{\frac{1}{2}+\frac{4\cdot Q}{\pi \cdot D^{2}\cdot L}}}$$
where *c* represents the sound velocity in the medium contained inside the capillary in cm/s, *D* represents the internal capillary diameter in cm, *Q* represents the sensor cavity volume in cm^3^ and *L* represents the capillary length in cm. It is important to note that the previous formula results from a theoretical calculation that predicts the natural frequency of a tube connected to a closed volume, in this case the sensor’s diaphragm. The sensor cavity volume is the slight depression between the diaphragm and the sensor body wall and is generally specified in the datasheets of all dynamic pressure sensors.

Considering *c* = 340 m/s, a minimum capillary length of 50 cm, a maximum capillary length of 400 cm, a capillary diameter of 5 mm and a sensor cavity volume of 0.2 cm^3^, Figure 16 represents the theoretical and experimental natural frequency of a pressure channel that includes the pressure sensor and the connecting pneumatic capillary with a length that varies between 50 and 400 cm with 17.5 cm increments.

It is important to note that without any capillary tube the natural frequency is about 250 Hz, which agrees with the datasheet specifications of the transducer under test.

## 6. Conclusions

This paper presents a low-cost and flexible calibrator protype that can be used to perform static and dynamic pressure calibrations, using periodic and aperiodic calibration methodologies. As well as for calibration, the prosed prototype can be used as an arbitrary waveform pressure generator. From the experimental results that were obtained, the prototype exhibited a static calibration error lower than 0.6% of the full-scale range, a dynamic periodic error lower than 2% for pressure signals whose frequency can range between 0 and 70 Hz and whose pressure amplitude can vary between 0 and 34.5 kPa, and a relative error lower than 4% using the aperiodic step calibration method. Regarding additional capabilities, the proposed prototype can generate any arbitrary pressure waveform that is previously defined by the user. This advantage is particularly interesting with respect to testing and calibrating pressure measurement devices for biomedical applications, such as NNS stimulation and digital sphygmomanometers, among others. Additional novelties and advantages of the proposed prototype include its remote pressure sensing capabilities that can be used to minimize calibration errors, such as the ones associated with capillary effects and pressure leakages, its portability and its low cost.

## Figures and Tables

**Figure 1 sensors-24-00629-f001:**
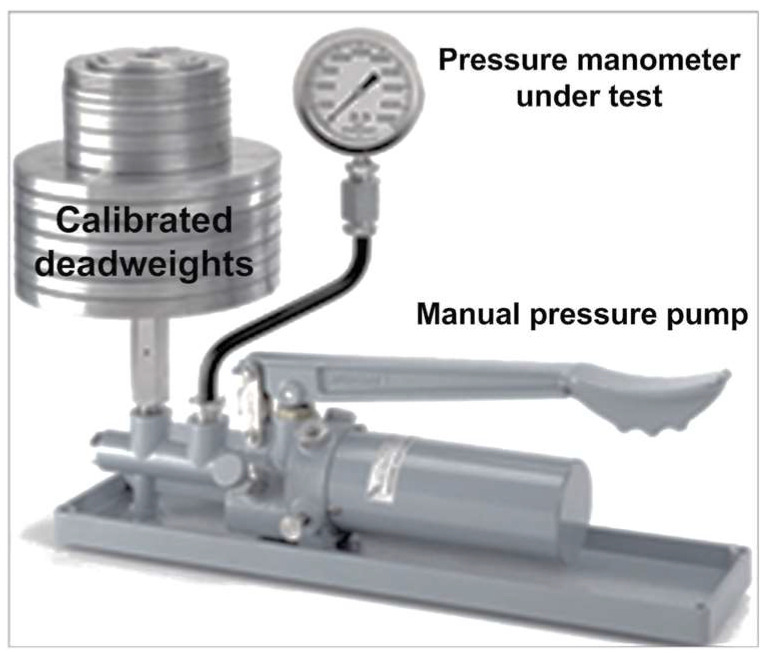
Deadweight tester for static manometer calibrations.

**Figure 2 sensors-24-00629-f002:**
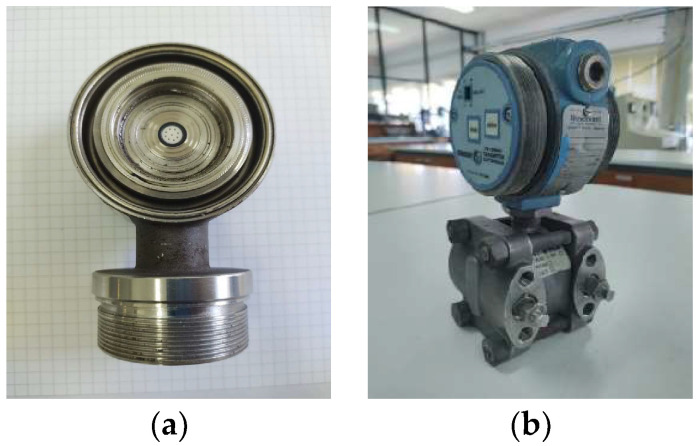
Differential pressure transmitter: (**a**) sensor and conditioning circuit parts; (**b**) d/P capacitive cell.

**Figure 3 sensors-24-00629-f003:**
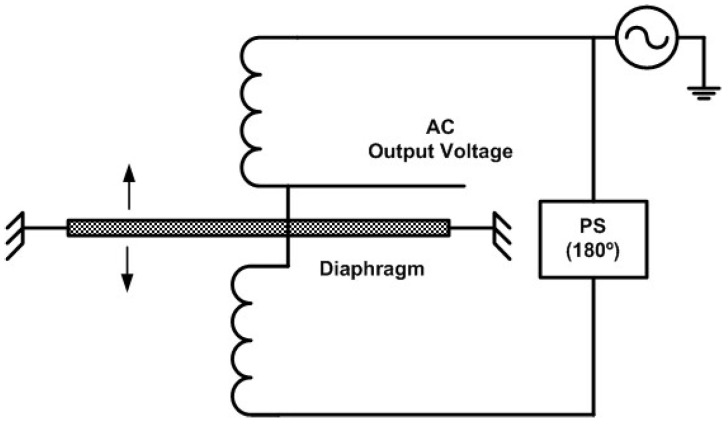
Schematic diagram of the measuring principle of a reluctance pressure transducer (PS—phase shifter).

**Figure 4 sensors-24-00629-f004:**
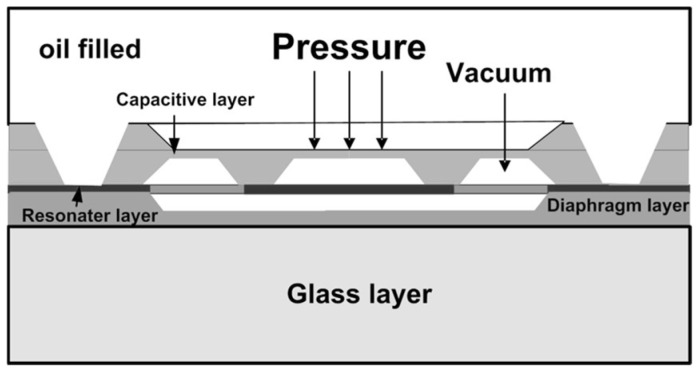
Cross section of a TERSP resonator.

**Figure 5 sensors-24-00629-f005:**
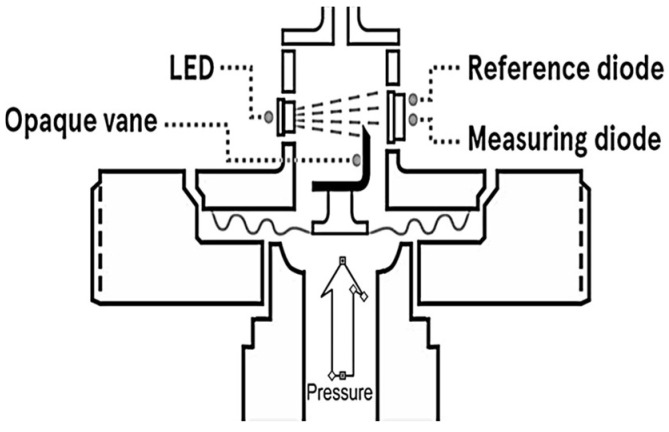
Possible arrangement of an optical pressure sensor.

**Figure 6 sensors-24-00629-f006:**
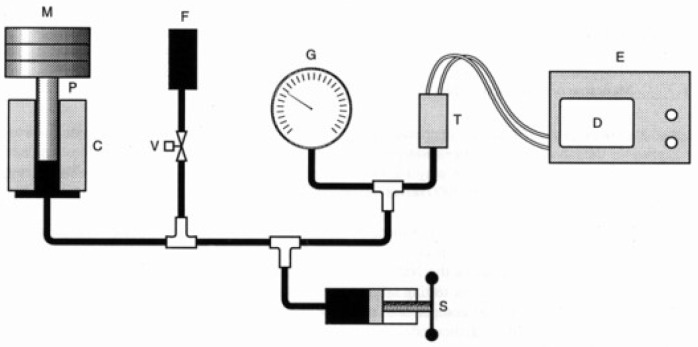
Schematic diagram of a typical deadweight tester (M—reference mass; P—piston; C—cylinder; V—manual valve; F—fluid reservoir; G—gauge to be calibrated; T—pressure transducer; S—screw pump; E—equipment (power supply with delivered current measurement capability); D—display).

**Figure 7 sensors-24-00629-f007:**
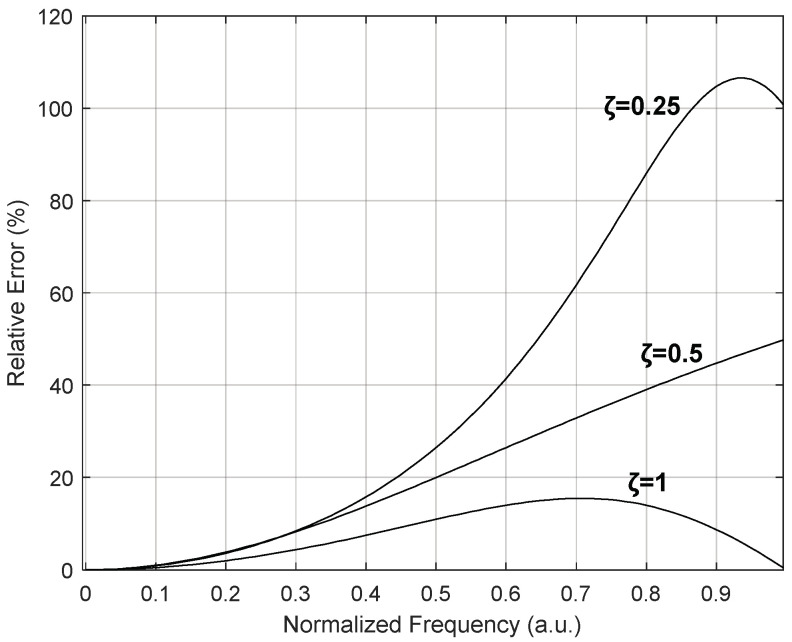
Relative measurement errors associated with using static calibration for pressure sensors that are working with time-variable pressures (ζ = 0.25; ζ = 0.5; ζ = 1).

**Figure 8 sensors-24-00629-f008:**
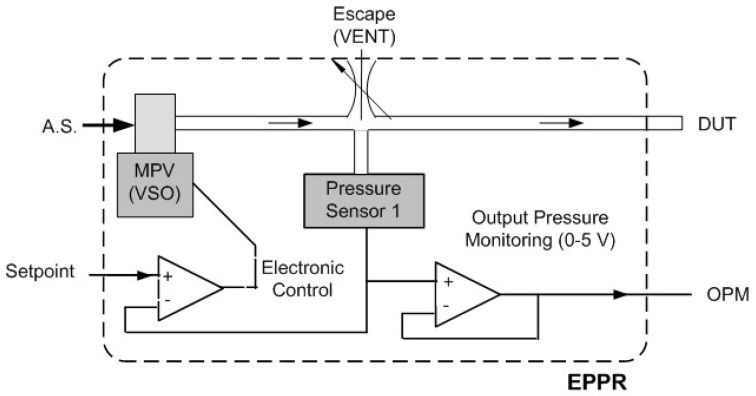
EPPR block diagram (EPPR—electropneumatic pressure regulator; MPV—miniature proportional valve; OPM—output pressure monitoring; A.S.—air supply).

**Figure 9 sensors-24-00629-f009:**
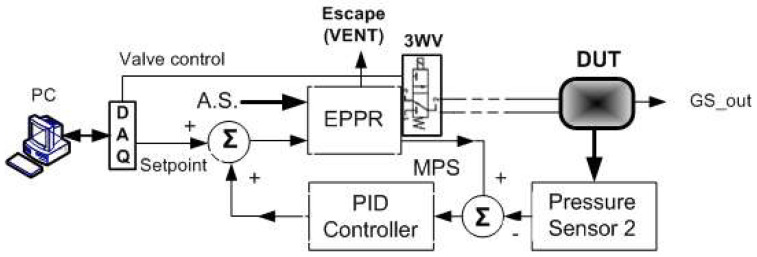
Calibration system block diagram (EPPR—electropneumatic pressure regulator; DAQ—data acquisition board; A.S.—air supply; 3WV—three-way solenoidal valve; DUT—device under test; S_out—DUT signal output).

**Figure 10 sensors-24-00629-f010:**
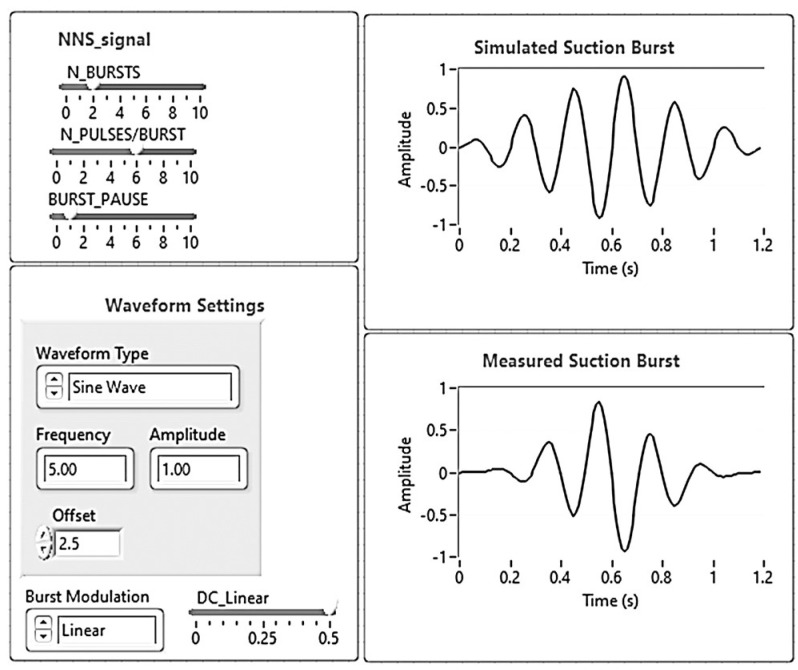
Front panel of the LabVIEW virtual instrument used to define the parameters of the synthesized NNS signal.

**Figure 11 sensors-24-00629-f011:**
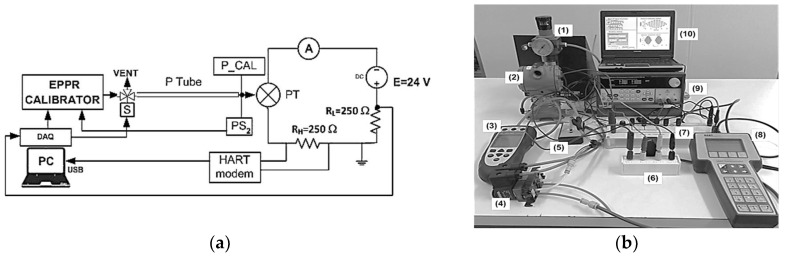
Experimental setup used for testing and calibration purposes: (**a**) EPPR—electro pneumatic pressure regulator; P Tube—pneumatic tube; PS—pressure sensor; PT—pressure transducer; A—amperemeter; RL—loop resistance; RH—HART communication resistor; E—loop voltage power source; S—solenoidal element of a three-way valve; (**b**) (1) pressure regulator; (2) pressure transmitter; (3) pressure calibrator; (4) three-way valve; (5) pressure sensor; (6) EPPR; (7) measuring chain module; (8) HART communicator; (9) power supply with current measurement capability; (10) laptop with data acquisition board.

**Figure 12 sensors-24-00629-f012:**
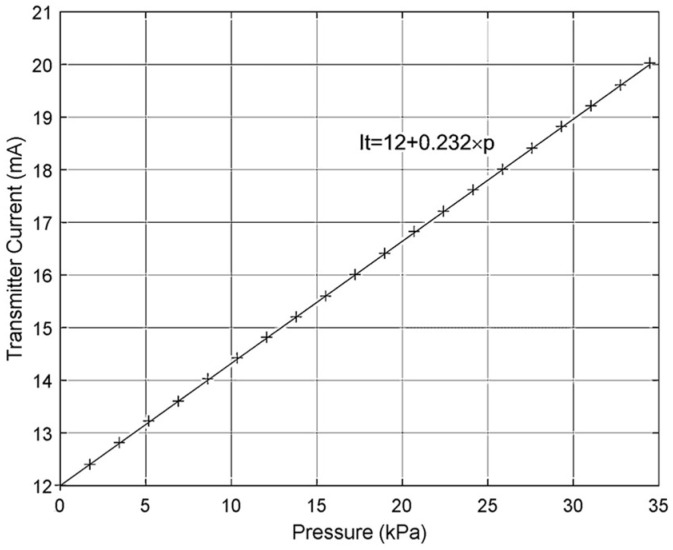
Static calibration results (theoretical values—continuous line; + symbols—experimental values).

**Figure 13 sensors-24-00629-f013:**
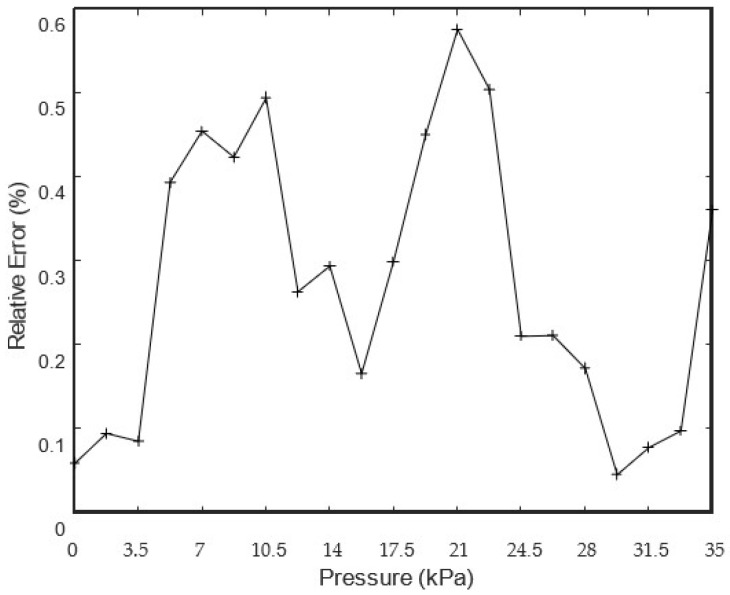
Relative error of the static calibration (interpolated errors—continuous lines; + symbols—experimental errors).

**Figure 14 sensors-24-00629-f014:**
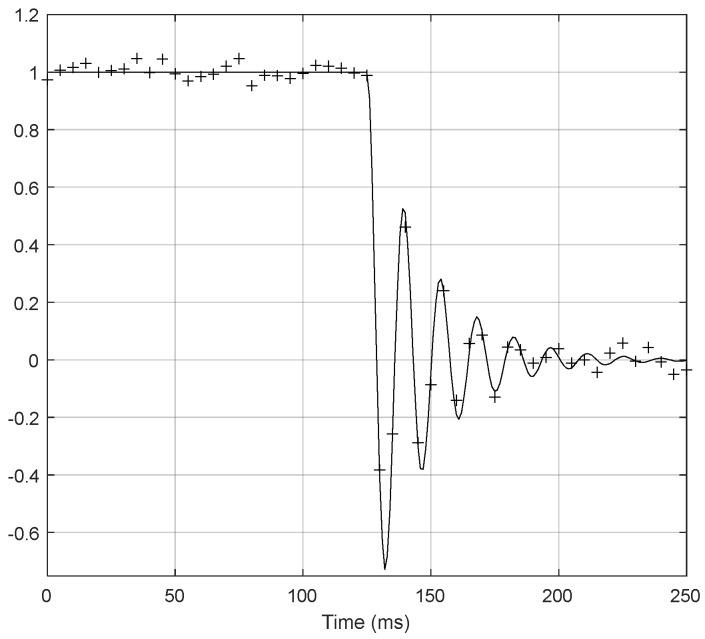
Dynamic calibration results (theoretical values—continuous line; + symbols—experimental values).

**Figure 15 sensors-24-00629-f015:**
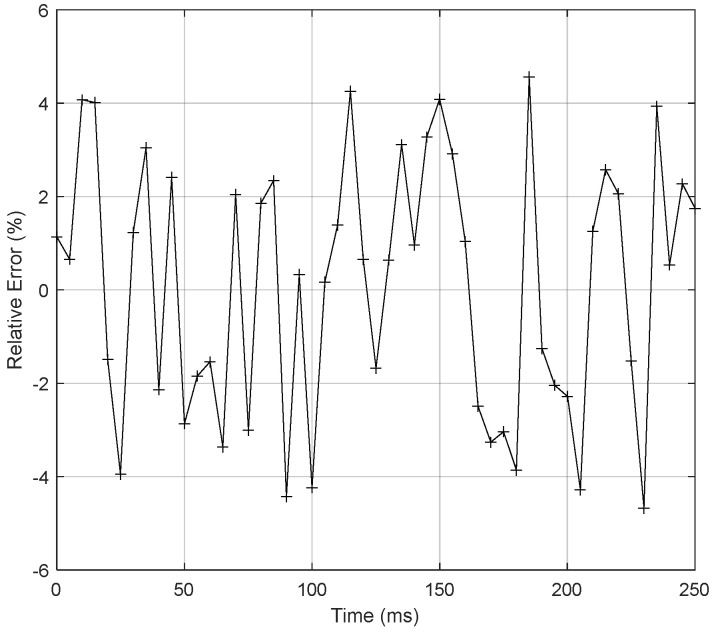
Relative error of the dynamic calibration (interpolated errors—continuous lines; + symbols—experimental errors).

**Figure 16 sensors-24-00629-f016:**
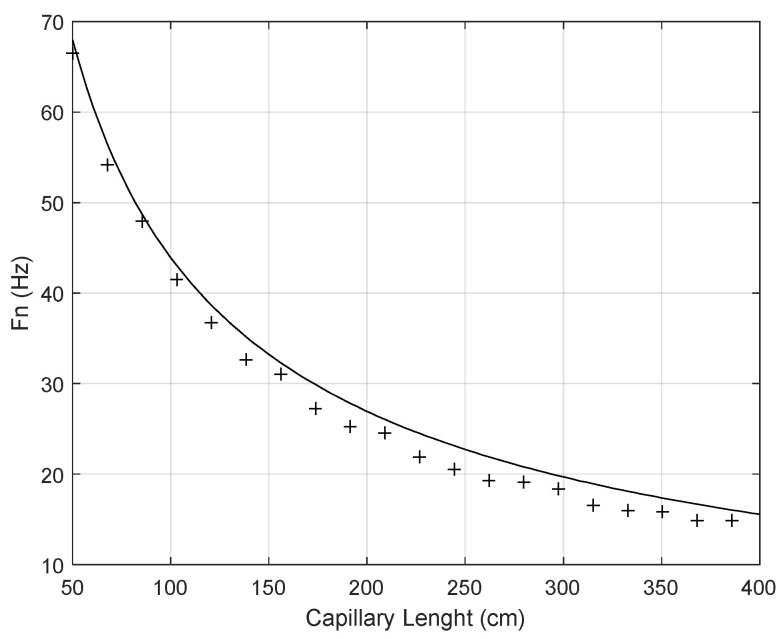
Natural frequency of a pressure channel that includes the pressure sensor and the connecting pneumatic capillary (theoretical values—continuous line; + symbols—experimental values).

**Table 1 sensors-24-00629-t001:** Comparative analysis of different solutions for pressure calibration purposes.

Technique	P&A_CC *	Flexibility (I + B + AW) *	Portability	RSCC *	Pressure Range (kPa)
Loudspeaker [16]	Y	Y	Y	N	1–10
Pistonphone [17]	N	N	Y	N	10–50
Tubing system [10]	Y	Y	N	N	20–30,000
Dropping mass [18]	N	N	N	N	700–100,000
Proposed prototype	Y	Y	Y	Y	0–35

* Abbreviations: P&A_CC—periodic and aperiodic calibration capabilities; I + B + AW— industrial, biomedical and arbitrary waveform; RSCC—remote sense compensation capability; Y—yes; N—no; L—low; M—medium; H—high.

## Data Availability

Data are contained within the article.

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
