# Peer review of "Pressure Sensors: Working Principles of Static and Dynamic Calibration"

_sensors, 2024, doi:10.3390/s24020629_

Round 1

Reviewer 1 Report

Comments and Suggestions for Authors

The article is about static and dynamic calibration methods for pressure sensors. The structure of the article is correct but I have many comments, which I will post below and ask the author to discuss them and make changes 

1 Standardize the unit of pressure - use a unit consistent with the SI system, i.e. Pascal, everywhere 

2 Line 33 you write "but there are some important exceptions where dynamic calibration is essential" -> you need to list these cases 

3 line 37 - 44 for the examples listed you need to provide a literature reference 

4 line 119 convert to Pa

5 line 89 in this paragraph you write about sensors using diaphragm - what is the measured diaphragm strain in your opinion ? This is usually done with strain gauges please explain 

6 line 123 in this paragraph about sensors using strain gauges what this strain gauge is mounted on (diaphragm ?) 

7 line 161 You write that there is no ADC and in the above mentioned there is?  In my opinion it depends on the technology if we have a capacitive sensor with a digital output then the converter is, if we have an analog output then there is no ADC. The same relationship will be for resistive sensors 

8 line 163 and on - the advantages are listed but their specific values should be given 

9 line 212 the pressure is mentioned twice, the electrical charge is missing

10 line 290 Figure 6 specify what kind of transducer and equipment 

11 lines 329 - 337 can you provide a formula  that would define the relationship between response time and damping. It would be useful 

12 line 448 A photo of the real calibration stand with appropriate descriptions should be added at the end of this chapter 

Also add information about the minimum and maximum pressure that can be generated and the minimum and maximum Burst frequency 

13 line 464 and 475 pressure should always be given in Pa

14 line 477 "RTD and thermocouple simulator." Why is the temperature sensor simulator used ? 

15 line 538 formulas should be numbered and reference should be made to the appropriate number 

16 The conclusion should be greatly expanded it should state the measurable parameters of the developed calibration bench. explain with what errors the calibration can be carried out. Necessarily refer to the literature and other scientific works on dynamic calibration of pressure sensors. Present in what the developed positions are better in what worse.

END

Reviewer 2 Report

Comments and Suggestions for Authors

sensors-2800568

Pressure Sensors: working principles, static and dynamic calibration 

·         There is a lack of novelties in the proposed work.

·         Lots of theory parts are included without reference. The history part of the sensor in the Introduction seems unnecessary.

·         A real Picture of an experimental setup should be included with this work (figure 11).

·         Figure 15 shows that relative errors are more than 4%, which is relatively high.

·         Please explain Figure 14 more clearly.

·         In figure 11, what are the typical values of Rh, RL, etc.?

·         It will be good to add the real picture of the Labview front panel. Figure 10.

·         It is better to list a comparison table to compare results with previous work.

·         The novelty of the work should be clearly highlighted (in the abstract as well as in the conclusions).

Comments on the Quality of English Language

Minor editing of English language required

Round 2

Reviewer 1 Report

Comments and Suggestions for Authors

The article has been corrected I recommend publication.

(Authors can pay attention to the correct formatting of fig 11 the photo should be more readable) 

Author Response

Please, see annexed file.

Reviewer 2 Report

Comments and Suggestions for Authors

Thank you for allowing me to revise the resubmitted manuscript " Pressure Sensors: working principles, static and dynamic calibration." The submitted manuscript and presented work is suitable for publishing in the Sensors, except for some minor revisions.

Minor revision:

1-Figure 11b, the picture of the experimental setup needs to be clarified. Components' names should be clearly shown.

2-In Table 1, it will be good to specify the pressure range. Also, why is the proposed work novel in comparison to other works?

  Comments on the Quality of English Language

Minor editing of English language required.

Author Response

Please, see annexed file.
